# Transcutaneous Detection of Intramural Microchips for Tracking the Migration of the Equine Large Colon: A Pilot Study

**DOI:** 10.3390/ani12233421

**Published:** 2022-12-05

**Authors:** Sara KT Steward, Hannah M. McKee, Allison M. Watson, Mo D. Salman, Diana M. Hassel

**Affiliations:** 1Department of Clinical Sciences, Colorado State University, Fort Collins, CO 80523, USA; 2Department of Microbiology, Immunology & Pathology, Colorado State University, Fort Collins, CO 80523, USA

**Keywords:** colic, horse, colonic displacement, microchip

## Abstract

**Simple Summary:**

Colic due to large colon displacement is one of the leading causes for equine hospitalization and surgery, yet there is not an adequate model to study the pathophysiology of this condition. The objective of this proof-of-concept study was to determine if subserosal implantation of bioinert microchips in the large intestine would be detectable by a receiver when the implanted microchips were adjacent to the body wall, thus identifying the location of the colon within the abdomen. The implantation of microchips with successful localization of various regions of the large colon was accomplished with no adverse effects identified. The future goal is to apply this safe and reliable model to monitor colonic movement in relation to various stimuli in a larger population of horses that can help identify environmental and dietary risk factors and preventative methods for colic due to large colon displacement.

**Abstract:**

Colic remains the number one cause of mortality in horses, and large colon displacement including colonic volvulus is one of the leading causes for equine hospitalization and surgery. Currently, there is not an adequate model to study the pathophysiology of this condition. The objective of this proof-of-concept study was to determine if subserosal implantation of bioinert microchips in the large intestine would be detectable by a RFID (radio-frequency identification) receiver when the implanted microchips were adjacent to the body wall, thus identifying the location of the colon within the abdomen. A horse with no history of gastrointestinal disease underwent a ventral midline celiotomy to implant twelve bioinert microchips into the subserosa at predetermined locations within the large colon and cecum. A RFID scanner was used to monitor the location of the colon via transcutaneous identification 1–3 times daily for a one-month period. Following humane euthanasia, a postmortem examination of the horse was performed to assess microchip implantation sites for migration and histologic assessment. Eleven out of the 12 implanted microchips were successfully identified transcutaneously at occurrences as high as 100%. Odds ratios were calculated for the likelihood of identifying each chip in a location different from its most common location. Microchips implanted into the subserosa of the equine large colon can be used as a means of identifying the approximate location of the equine large colon via transcutaneous identification with an RFID scanner.

## 1. Introduction

Colic in the horse is a costly disease for horse owners and professionals alike. Horses are hindgut fermenters [1] where they rely on their cecum and large colon to utilize roughage sources for energy. The equine large colon is made up of four anatomic segments. The right and left ventral colons followed by the left and right dorsal colons. While the base of the right colon is attached to the right dorsal body wall, the left colon makes up what is known as the pelvic flexure, which is freely movable throughout the abdomen. This unique anatomic structure contributes to the large colon’s ability to mix and ferment fibrous material. It is also a very common location for obstruction resulting in colic signs in the horse [2,3]. Large colon displacement makes up approximately 16% of colics seen at referral hospitals, with 24–36% of these cases needing surgical intervention [2,3].

While treatment for large colon displacement has been well described in the literature [3,4,5,6,7], the mechanisms and risk factors behind colonic displacement have yet to be determined. This is largely due to the difficulty of distinguishing the location of the large colon within the abdomen at any given timepoint. Ultrasonography may be able to infer the presence of a left or right dorsal displacement [8,9,10]; however, the exact location of the pelvic flexure at any given moment is impossible to accurately predict. Diagnostics such as radiography or computed tomography are not readily applicable to the equine abdomen due to its large size and volume [11]. While there have also been methods described to help prevent displacement of the large colon such as ablation of the nephrosplenic space [12,13] and colopexy [14,15,16], these procedures do not address the underlying cause of colonic displacement. In the case of nephrosplenic space ablation, the colon is still able to displace to other locations in the abdomen. With colopexy, the immobility of the colon may lead to signs of colic or even the catastrophic rupture of the colon [15]. The development of this model to characterize the location of the large colon is critical to recognizing the mechanisms of colonic displacement as well as the effects of changes in management, feed, gastric distension, and various activities on the movement of the large colon.

The aim of this pilot study was to develop a model for tracking the migration of the equine large colon within the abdomen. Secondary aims were to verify that the microchips did not induce significant injury at the sites of implantation and that the microchips would not migrate from the original site of implantation. We hypothesized that this model would be simple in application, that the microchips would remain in place and inert, and that transcutaneous identification would be feasible for the duration of the study. We also hypothesized that the locations most distant from the body wall attachments (pelvic flexure, cecal apex) would be found in more highly variable locations than those closer to the body wall attachments (right dorsal colon, right ventral colon).

## 2. Materials and Methods

### 2.1. Animals

One adult horse donated for reasons unrelated to the gastrointestinal tract and with no historical gastrointestinal disease or colic was used for this exploratory experiment. All procedures were approved by the Institutional Animal Care and Use Committee at Colorado State University with the implementation of the NIH guidelines for the care and use of animals. 

### 2.2. Methods

A grid pattern was created on the horse with clipped lines on the horse’s body using a #40 clipper blade at the midpoint between the xiphoid and the caudal aspect of the tuber coxae. Two additional vertical gridlines bisected each vertical segment. Likewise, 2 horizontal gridlines were marked at the level of the caudodistal tuber coxae and the point of the elbow, with the final horizontal gridline along the ventral midline. The gridlines resulted in 15 static areas on each side of the horse (R1–15, L1–15) as well as 10 static areas on the ventral aspect of the horse (V1–10) (Figure 1).

The microchips were surgically implanted via ventral midline celiotomy under general anesthesia and in dorsal recumbency (Figure 2). The routine protocol for ventral midline surgical approach was performed. The horse was fasted for 12 h prior to the procedure. The large colon and cecum were exteriorized from the abdomen and assessed for any gross abnormalities or variations from normal prior to implantation of the microchips. The 12-mm microchips were then implanted subserosally using the implantation needle included from the manufacturer (P/N AVID2328; Avid Identification Systems, Inc., Norco, CA, USA). 

The specific implantation sites were as follows: pelvic flexure (antimesenteric), left ventral colon (dorsal and ventral bands, measured at a midpoint between the pelvic flexure and the sternal flexure), left dorsal colon (antimesenteric, at the same level as the left ventral colon microchips), diaphragmatic flexure (antimesenteric), sternal flexure (ventral band), right dorsal colon (antimesenteric, measured at a midpoint between the cecocolic ligament and the sternal flexure), right ventral colon (ventral band, measured at the same level as the right dorsal colon microchip), and cecum (lateral band midway between cupula and apex, medial band at the same level as the lateral band microchip, dorsal band at the same level as the other two mid-body microchips, and apex). A total of 12 microchips were implanted (Figure 3).

The specific identification number of each microchip was recorded and confirmed with a chip scanner (Transponder Reader, iMax Black Label RID by DataMars SA, Lugano, Switzerland) for each site at the time of implantation. Following implantation, the microchips were oversewn with 0-glycomer 631 (Biosyn, Medtronic, Minneapolis MN, USA) in a cruciate pattern to prevent the movement of the microchip from the implant site. The cecum and pelvic flexure were replaced in their correct anatomical locations within the abdomen and the abdominal wall was closed in a routine fashion. No immediate complications were encountered and the horse recovered from the procedure routinely.

Routine perioperative care was provided, including flunixin meglumine (1.1 mg/kg IV q12hrs for five days), procaine penicillin G (22,000 units/kg IM once, administered under general anesthesia), and gentamicin (6.6 mg/kg IV once). The horse was housed in a stall for one week following surgical implantation and refed slowly over the first 12 h on a grass and alfalfa hay mixture. She was then offered hay ad libitum. Following the first week, she was moved to a small paddock. The horse was fed 6 flakes of a grass and alfalfa combination once daily in the morning. Microchip locations were recorded one to three times daily. The timing of the microchip location assessment in relation to feeding was recorded: during a fasted state (prior to morning feeding), during feeding, and 2–4 h following feeding. The microchip recordings generally took place between 1–4 h apart, based on fasted, feeding, or post-feeding timelines. The horse was maintained for 32 days prior to humane euthanasia (pentobarbital 86 mg/kg IV) at the end of the study. No complications arose from the implantations or surgical site for the duration of the study.

Following humane euthanasia 32 days post microchip implantation, post-mortem examination of large colon and cecal tissues was performed. The location of the microchips at the time of postmortem examination was recorded and tissue samples were collected from multiple sites of implantation, placed in 10% neutral buffered formalin, and processed routinely for a histopathologic analysis.

### 2.3. Statistical Analysis

The unit of the study was an implanted microchip by each grid segment. Descriptive statistics were used to provide the frequency with which each microchip was identified by each implanted site. The frequency of the detection of each microchip by each grid segment across time was assessed for its statistical significance using an Odds Ratio as an estimate to compare among the various segments. Ultimately, the likelihood of a chip to deviate from its most common location was tested. A description of the movement of each microchip across the time of the study was presented using bar charts by each grid segment. As this was a proof-of-concept pilot study, a sample size calculation was not performed without the ability to make inferences from the generated data. The calculation of the frequency and odds ratio were performed using an available epidemiological tool—Openepi (http://www.openepi.com/Menu/OE_Menu.htm (accessed on 7 July 2022)).

## 3. Results

### 3.1. Microchip Identification

The pilot study was able to be performed as anticipated with no complications in the implantation process. The microchips were transcutaneously identifiable via a microchip scanner for the duration of the study. One microchip (cecal apex) was identified in manure on the 8th day of the study. The degree of movement was highly variable among implant locations. Some chips moved minimally (e.g., sternal flexure found in the same grid location much of the time) and others moved commonly (e.g., pelvic flexure could be detected on the opposite side of the abdomen within a 2-h window). 

Following euthanasia and prior to the post-mortem examination, a single timepoint scan was performed in the left lateral recumbency and then dorsal recumbency. The microchip that was implanted in the right dorsal colon, which had previously never been identified, was located in the right flank when the horse was positioned in dorsal recumbency on the hoist but was not identified in left lateral recumbency. At post-mortem examination, 10 of the twelve microchips (83%) were identified in their originally implanted locations. The microchip that was implanted in the dorsal band of the left ventral colon was not identified at the postmortem examination and was presumed to have been lost intraluminally in a similar manner to the cecal apex microchip.

### 3.2. Necropsy and Histopathology

The gross necropsy findings demonstrated variable changes surrounding microchip implant sites from no apparent gross abnormalities (Figure 4A) to evidence of mild focal inflammatory changes (Figure 4B).

Histopathologic findings at colonic implant sites were characterized primarily by mild to moderate eosinophilic and lymphoplasmacytic colitis with mild submucosal edema and regional serosal granulation tissue (Figure 5).

### 3.3. Frequency of Microchip Identification

The most commonly identified microchips throughout the study were the pelvic flexure, the ventral band of the left ventral colon, the sternal flexure, the diaphragmatic flexure, the right ventral colon, and the lateral band of the cecum (Figure 6A–E). The most common location for identification of the pelvic flexure microchip was the L10 segment, where it was identified 19/80 times (23.8% of scans). The pelvic flexure was also identified on the right side of the abdomen 24/80 times (30%), in the ventral abdomen 10/80 times (12.5%), and the left side 25/80 times (31.3%). In total, the pelvic flexure microchip was identified 55/80 times (68.8%) (Figure 6A). Interestingly, the pelvic flexure was identified in the cranioventral right abdomen immediately following ventral midline celiotomy, where the pelvic flexure was placed in the left caudodorsal abdomen (Figure 7). The overall frequency of identification of all microchips is reported in Table 1.

## 4. Discussion

This pilot study successfully proved the ability to implant microchips subserosally in the large intestine of a living horse. The microchips had good retention, remained at the implantation sites 83% of the time, and induced minimal localized inflammation. Those that were lost, were lost intraluminally and did not cause complications to the animal. The authors believe that those microchips that were lost were likely implanted at an inappropriate subserosal depth, thus predisposing them to enter the lumen of the intestine transmucosally. Therefore, the retention percentage of the microchips would likely increase as the implanter’s learning curve improved. 

The microchips that were most commonly identified (PF, LVC, SF, RVC, Cecum LB) were all implanted in locations of the colon and cecum that were expected to be frequently adjacent to the body wall based on historically described anatomy. Of these, the chips that had the most variance in their identified segment locations were those of the implantation locations expected to be most mobile in the abdomen (pelvic flexure, left ventral colon, and lateral band of the cecum). Conversely, the sites that were commonly identified in less variable locations (RVC, SF) were associated with anatomic locations expected to migrate less due to their proximity to body wall attachments. 

The estimated depth of the detection of the microchips was approximately 15 cm, inclusive of the body wall, based on a 15 cm maximum diameter of detection externally. This is consistent with preliminary work performed on a necropsy specimen where a microchip attached to the right ventral colon was detectable from the external body wall only when adjacent to it. When the colon was manipulated 180 degrees, the chip could no longer be identified.

Throughout the study, the subject exhibited normal equine behavior with no signs of gastrointestinal distress or colic signs. Despite this, areas of the colon were identified in locations aberrant from anatomical description and potentially even consistent with subclinical displacement of the large colon and cecum. This substantiates the ability of the colon to move significantly within the abdomen in the absence of clinical signs of colic.

Because this migratory movement was able to be successfully logged, it is the hope of the authors that this model could be used to investigate risk factors, pathogens, or inciting causes of pathologic displacement of the equine large intestine. Implementation of this model to investigate the role of feed changes, management changes, rolling, physical activity, intestinal microbiota, and gastric distension on colonic movement may provide valuable insights into the causes and methods of prevention of colonic displacement and volvulus. For example, it has been proposed that gastric distension or impaction may alter colonic position and predispose the horse to large colon volvulus [17,18]. Using this model, this theory can be investigated and may positively influence methods of enteral fluid therapy for horses with mild forms of colic.

Further studies with greater numbers of horses, a longer study duration, and more controlled variables are indicated to further substantiate this model. The implantation process was well tolerated and was verified with post-mortem histopathology, thus negating the need for further terminal proof of concept studies in the future. Research horses used in future models could be kept indefinitely with implanted microchips in order to track migration over a long period of time. The precedent for research herds of this nature include teaching horses in university setting or blood/rumen fluid donation animals.

By completing this pilot study, the authors hope to provide a tool for further investigation into the pathogenesis of large colon displacements and volvulus in the horse.

## 5. Conclusions

This proof-of-concept study demonstrated successful localization of various portions of the equine cecum and large colon during a variety of fed states without clinically significant complications observed. Normal movement of the cecum and colon, particularly of the pelvic flexure and adjacent regions was readily documented. This model will prove to be useful in future studies assessing the impact of various stimuli on colonic movement or displacement with the long-term goal of reducing risk for this common form of colic in horses. 

## Figures and Tables

**Figure 1 animals-12-03421-f001:**
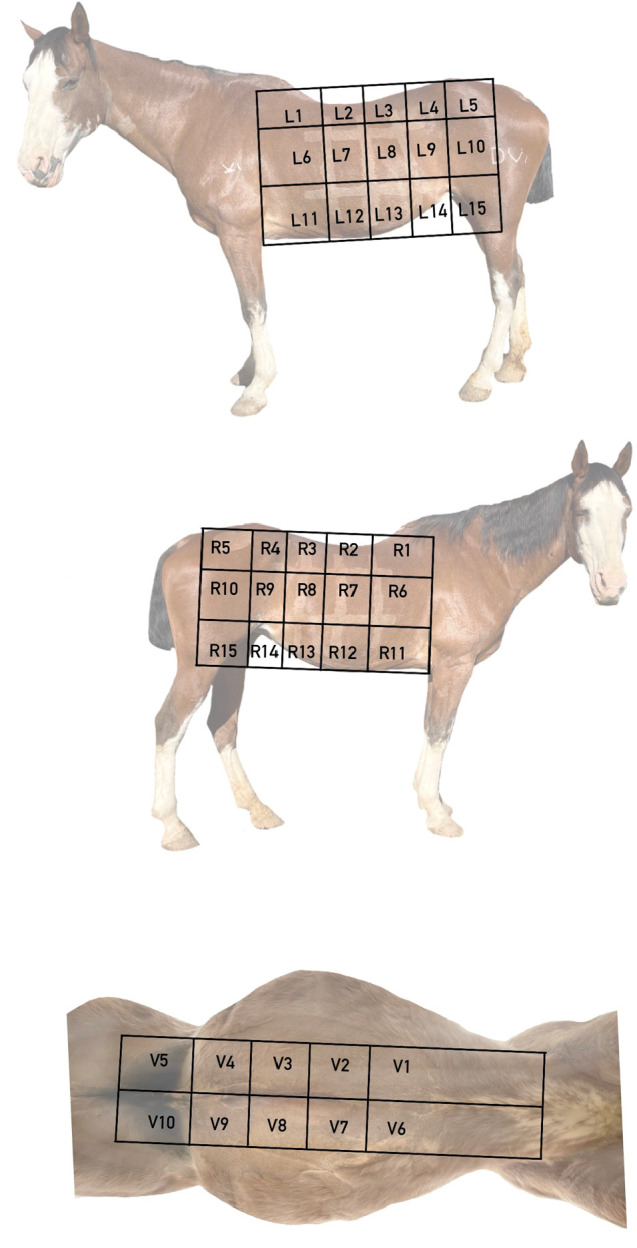
Gridlines dividing the ventral midline and left and right body walls into 40 discreet sections for microchip identification, denoted using clipped lines of body hair.

**Figure 2 animals-12-03421-f002:**
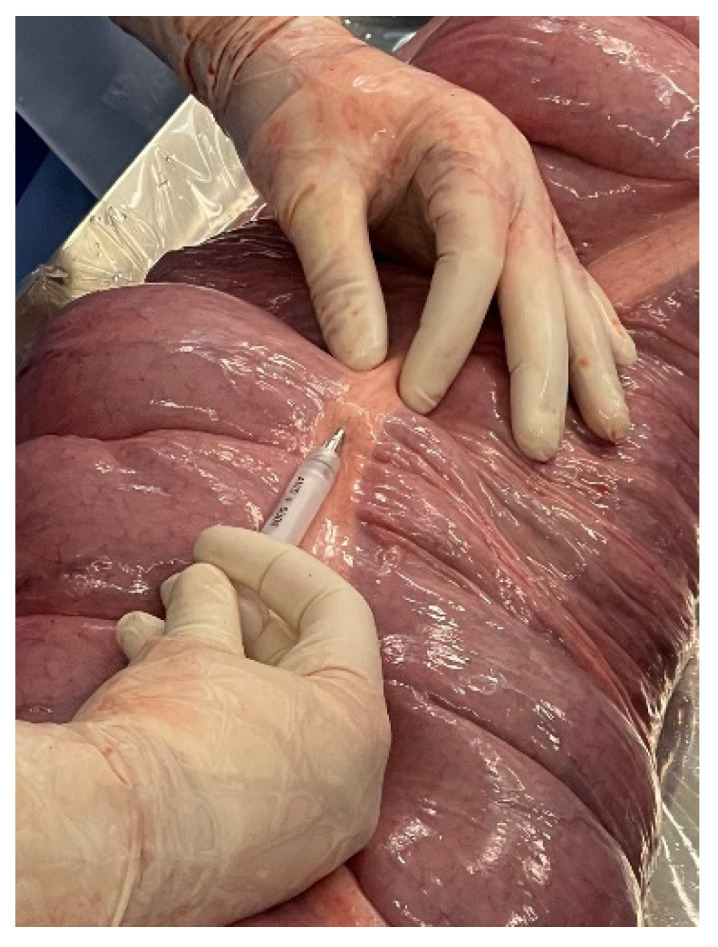
Surgical implantation of microchip into submucosal tissue of the ventral colon. A single cruciate suture was applied incorporating the implanted microchip to prevent migration from the site of implantation.

**Figure 3 animals-12-03421-f003:**
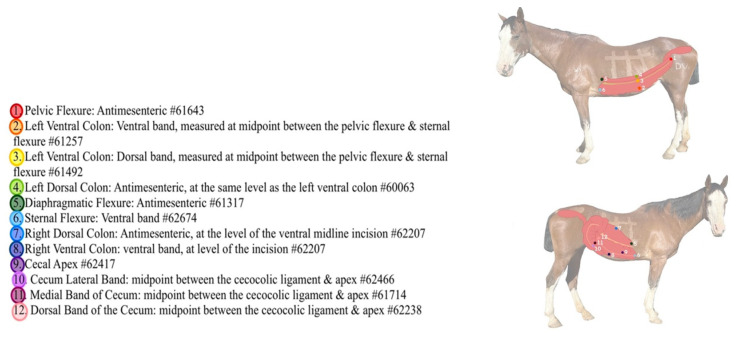
Color-coded key for locations of implanted microchips in the large colon and cecum.

**Figure 4 animals-12-03421-f004:**
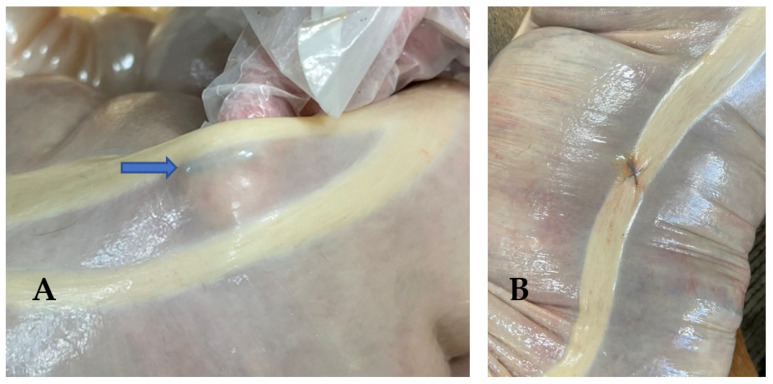
Gross pathologic findings at sites of microchip implants demonstrating variable responses to the implant from no apparent reaction (**A**) at the ventral band of the right ventral colon (site #8), to mild focal inflammatory changes (**B**) at the sternal flexure implant location (site #6). The blue arrow identifies the indwelling microchip.

**Figure 5 animals-12-03421-f005:**
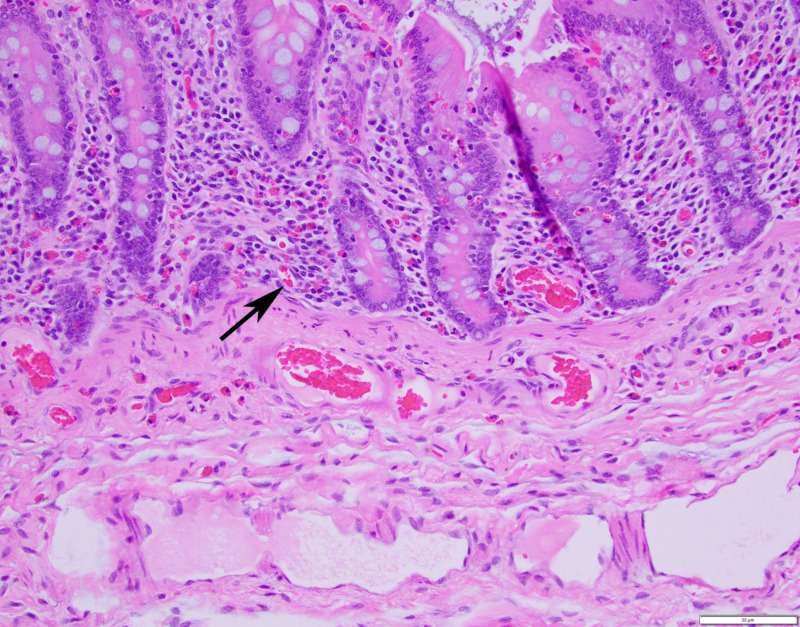
Histopathologic specimen of pelvic flexure mucosa and submucosa at the implant site (200×) demonstrating mild to moderate eosinophilic and lymphoplasmacytic colitis (arrow) and submucosal edema. Scale is 50 μm.

**Figure 6 animals-12-03421-f006:**
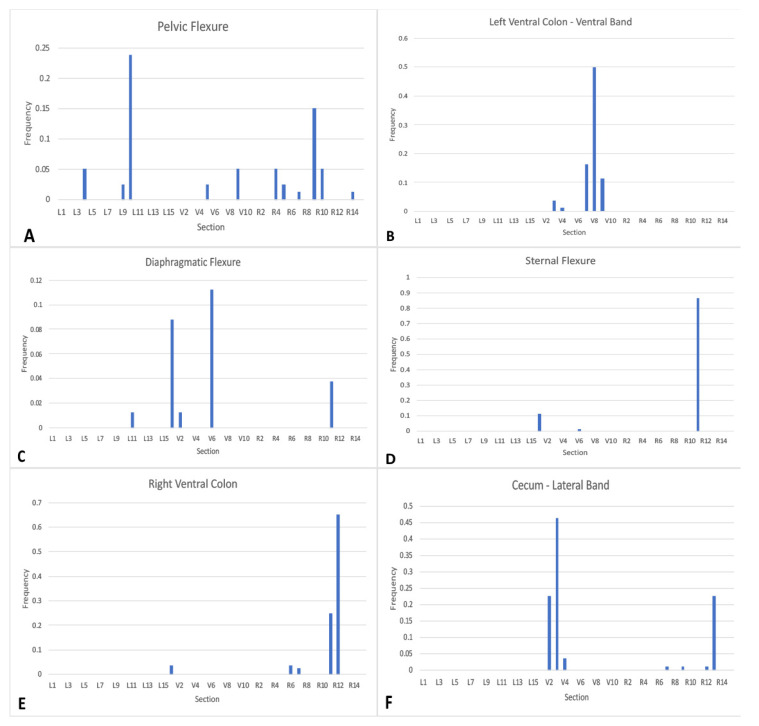
Bar graphs demonstrating the location and frequency of identification of microchips implanted in the pelvic flexure (**A**), the ventral band of the left ventral colon (**B**), the diaphragmatic flexure (**C**), the sternal flexure (**D**), the right ventral colon (**E**), and the lateral band of the cecum (**F**).

**Figure 7 animals-12-03421-f007:**
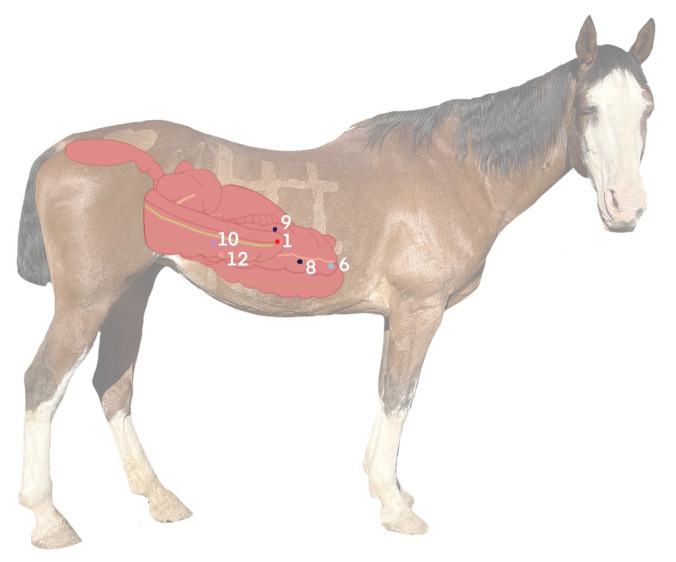
Graphical representation of the suspected position of the pelvic flexure (1) in the immediate post-operative period, characteristic of a form of right dorsal displacement of the colon. Notably, the pelvic flexure corrected its position to a “normal” left-sided position when scanned the following morning. Refer to Figure 3 for the color-coded key of implanted microchips.

**Table 1 animals-12-03421-t001:** The total frequency of identification and most commonly identified location of the nine microchips ^+^ that were identified throughout the duration of the study at 80 time points. PF = pelvic flexure, LVC = left ventral colon, LDC = left dorsal colon, DF = diaphragmatic flexure, SF = sternal flexure, RVC = right ventral colon, Cecum LB = lateral band, Cecum MB = medial band, Cecum DB = dorsal band.

Microchip	PF	LVC	LDC	DF	SF	RVC	Cecum LB	Cecum MB	Cecum DB
ID Frequency (%)	69	83	13	26	99	100	99	8	11
Location with its frequency as reference (%)	L10 24	V8 50	L13 6	V6 6	R11 86	R12 65	V3 46	V4 3	R13/14 * 4
Odds Ratio **	0.10	1.0	0.12	0.12	5.96	1.84	5.96	0.01	0.007
95% Confidence Interval	0.03, 0.29	0.02, 0.62	0.02, 0.62	0.02, 0.62	0.146, 244.0	0.184, 18.5	0.14, 244	0.0002, 0.07	0.0002, 0.05

^+^ Excluding those lost and the single microchip that was never identified despite confirmation of placement at postmortem examination. * The dorsal band of the cecum was identified equally in R13 and R14, each three times. ** The odds ratio was calculated based on the likelihood to identify the microchip in a location different from the most common location of that microchip. The reference Odds Ratio for each microchip (OR = 1) is the corresponding most frequent location from row 2 of this table.

## Data Availability

Not applicable.

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
