# Peer review of "Transcutaneous Detection of Intramural Microchips for Tracking the Migration of the Equine Large Colon: A Pilot Study"

_animals, 2022, doi:10.3390/ani12233421_

Round 1

Reviewer 1 Report

the authors describe a pilot study to detect the position of the large colon in a live healthy horse, after intramural application of microchips that could be detected transcutaneously

the results suggest that the technique applied is feasible and reliable

further studies could investigate the displacements of the large colon in a population of horses related to various predisposing factors, using the same technique

this data would be very useful for the scientific community, equine practitioners, and horse owners

I would suggest minor revisions:

Title: Transcutaneous detection (or evaluation) of intramural microchips ...

Introduction: line 65 - in situ, the migration is not "in situ" but within the abdomen

Author Response

Thank you for your review and suggestions for improvement.  We have implemented the requested changes along with several other improvements in response to reviewers comments.

the authors describe a pilot study to detect the position of the large colon in a live healthy horse, after intramural application of microchips that could be detected transcutaneously

the results suggest that the technique applied is feasible and reliable

further studies could investigate the displacements of the large colon in a population of horses related to various predisposing factors, using the same technique

this data would be very useful for the scientific community, equine practitioners, and horse owners

I would suggest minor revisions:

Title: Transcutaneous detection (or evaluation) of intramural microchips ...

Thank you for the suggested change. Title was changed.

Introduction: line 65 - in situ, the migration is not "in situ" but within the abdomen

Changed as requested.

Reviewer 2 Report

General comments:

This study approaches a new model in horses to monitor the colonic movements/displacement through the use of microchip implementation into the subserosa of large intestine in one animal.

The experimental study is well designed, and clear to read. Nerveless, in my opinion, the analysis and presentation of the results can be improved turning more easily the discussion of the findings; please find suggestions as specific comments for a moderate revision. Despite the above mentioned, I have two questions/curiosities about your experiment which were not approached: 1) What is the timeline and degree of movements? Can be the microchips implemented by laparoscopy?

Specific comments:

L114: I suggest to insert the chip number in the horse silhouettes.

L174: Please add identify the localization of the implant in Fig.4A, and insert the colonic local of both sites in the legend.

L176: What is the magnification of the slide?

L196: Here, the same problem using colors as identification of the microchips. The use of number chip will be more useful.

L182-187: It was stated in M&M that the frequency identification in each segment was evaluated by the Chi-Square test, but the significances were not reported.

L200: In Table 1, the odds ratio and 95%IC were reported but their significance was not tested. As example, according to the 95%IC the odds ratio for SF, RVC and Cecum LB seems non-significant. I suggest to add “as reference” in row 2 and “95%” in row 4.

L213: “…altered…”?

L221-228: Can you report what is the normal depth (or its interval) which the microchip scanner can read the microchips?

L256-257: Your study detected the (temporary) displacement of colon/cecum as you define as movement. Nevertheless, you only consider the frequency of detection and not the timeline across the whole experiment and the degree of movements were not evaluated. So, we know that the movement occurred, but not the distance and duration. I think that one/two paragraphs addressing the main advantage and limitation of this new method inserted in the discussion will be appreciate by the readers.   

Author Response

This study approaches a new model in horses to monitor the colonic movements/displacement through the use of microchip implementation into the subserosa of large intestine in one animal.

The experimental study is well designed, and clear to read. Nerveless, in my opinion, the analysis and presentation of the results can be improved turning more easily the discussion of the findings; please find suggestions as specific comments for a moderate revision. Despite the above mentioned, I have two questions/curiosities about your experiment which were not approached: 1) What is the timeline and degree of movements? Can be the microchips implemented by laparoscopy?

We have attempted to provide improved clarity by addressing all suggestions.  Thank you for your useful suggestions that have improved the quality of this manuscript. 

Regarding timeline, the scans took place daily over 30 days and typically occurred between 1-4 hours apart based on fasted, feeding or post-feeding timelines.  Degree of movement was highly variable among implant locations.  Some chips moved minimally (e.g. sternal flexure found in the same grid location much of the time) and others moved commonly (e.g. pelvic flexure could be detected on the opposite side of the abdomen within a 2-hour window).  This information was added to the methods, results and discussion as it may be of interest to readers.  We also created a website with a video that has a nice animation to demonstrate how the microchips would move in relationship to time: https://hannahmmckee.wixsite.com/migratingmicrochips.  We added a link to this in the results, but I am uncertain if this will need to be provided separately as supplemental material.

Regarding laparoscopic implantation, I think this is possible, but it would be challenging and perhaps impossible to accurately implant chips at all locations simply due to the need to access both dorsal and ventral aspects of the large colon and cecum.  Even with fasting, it is unlikely the colon could be adequately manipulated to access all sites of implantation.  That said, if a few select sites were targeted that were accessible via ventral approach (dorsal recumbency) or dorsal approach (standing laparoscopy), it may be feasible.

Specific comments:

L114: I suggest to insert the chip number in the horse silhouettes.

Chip numbers were added to both Figure 3 and Figure 7.

L174: Please add identify the localization of the implant in Fig.4A, and insert the colonic local of both sites in the legend.

Done!

L176: What is the magnification of the slide?

200X (bar is 50um).  Added to figure legend.

L196: Here, the same problem using colors as identification of the microchips. The use of number chip will be more useful.  Done.

L182-187: It was stated in M&M that the frequency identification in each segment was evaluated by the Chi-Square test, but the significances were not reported.

The lines in question now read "Frequency of the detection of each microchip by each grid segment across time was assessed for its statistical significance using Odds Ratio as an estimate to compare among the various segments" - The Chi-square test had been used in an earlier revision of the manuscript, but was decided to be an inaccurate depiction of the statistics in this study. We therefore decided to use only Odds Ratio for comparison between segments. The included verbiage, now deleted, was an editorial error on the authors' part when submitting. Thank you for recognizing this error.

L200: In Table 1, the odds ratio and 95%IC were reported but their significance was not tested. As example, according to the 95%IC the odds ratio for SF, RVC and Cecum LB seems non-significant. I suggest to add “as reference” in row 2 and “95%” in row 4.

Suggested edits made to the table.  Chi-square to provide P-values not performed (see response above).

L213: “…altered…”?

I am unclear as to the desired change as we felt the chips did not alter normal GI physiology.  I removed the sentence and added “and induced minimal localized inflammation” in the following sentence.

L221-228: Can you report what is the normal depth (or its interval) which the microchip scanner can read the microchips?

The following paragraph was added in response to your request:  Estimated depth of detection of microchips was approximately 15cm, inclusive of the body wall, based on a 15cm maximum diameter of detection externally. This is consistent with preliminary work performed on a necropsy specimen where a microchip attached to the right ventral colon was detectable from the external body wall only when adjacent to it. When the colon was manipulated 180 degrees, the chip could no longer be identified.

L256-257: Your study detected the (temporary) displacement of colon/cecum as you define as movement. Nevertheless, you only consider the frequency of detection and not the timeline across the whole experiment and the degree of movements were not evaluated. So, we know that the movement occurred, but not the distance and duration. I think that one/two paragraphs addressing the main advantage and limitation of this new method inserted in the discussion will be appreciate by the readers.  

Some additional data was added to the methods and results under “3.1. Microchip Identification” to add additional data regarding timeline and movement of chips.  Also, a link to the webpage with an animated depiction of movement of the colon was provided.

Round 2

Reviewer 2 Report

Dear authors,

thanks for providing this revised version. All comments/sugestions were considered and, in opinion, this version is able to be accepted by Animals.

Congratulation for your site! May be it permanence (specially the microchip movements) can be ensured if hosted by MDPI.